## [Decision Letter · Decision Letter 0]

17 Apr 2025

PCOMPBIOL-D-25-00322

Mechanistic computational modeling of sFLT1 secretion dynamics

PLOS Computational Biology

Dear Dr. Amy Gill,

Thank you for submitting your manuscript to PLOS Computational Biology. After careful consideration, we feel that it has merit but does not fully meet PLOS Computational Biology's publication criteria as it currently stands. Therefore, we invite you to submit a revised version of the manuscript that addresses the points raised during the review process.

Please submit your revised manuscript within 30 days. If you will need more time than this to complete your revisions, please reply to this message or contact the journal office at ploscompbiol@plos.org. Please include the following items when submitting your revised manuscript:

We look forward to receiving your revised manuscript.

Kind regards,

Mariko Okada

Academic Editor

PLOS Computational Biology

Pedro Mendes

Section Editor

PLOS Computational Biology

**Journal Requirements:**

Potential Copyright Issues:

i) Figures 1, and 2: We note that the figures are created through BioRender. Please confirm that you hold a Premium account and provide a pdf copy of the CC BY 4.0 License as provided by BioRender. For instructions on how to generate a CC BY 4.0 license for your figure, please see the guidelines here: https://help.biorender.com/hc/en-gb/articles/21282341238045-Publishing-in-open-access-resources. 

If you are using the free assets from BioRender, we are unable to publish these images as they are licenced under a stricter licence than CC BY 4.0. In this case we ask you to remove the BioRender images and replace them with open source alternatives.

See these open source resources you may use to replace images / clip-art:

- https://bioart.niaid.nih.gov/ 

- https://bioicons.com/

- https://healthicons.org/ 

- https://scidraw.io/

- https://reactome.org/icon-lib

- https://www.phylopic.org/images

5) Please provide a completed 'Competing Interests' statement, including any COIs declared by your co-authors. If you have no competing interests to declare, please state "The authors have declared that no competing interests exist".

**Reviewers' comments:**

Reviewer's Responses to Questions

Reviewer #1: The authors describe 8 mechanistic models for the production and secretion of sFLT1 involved in VEGF signaling. Comparing these models to existing experimental sFLT1 data, the authors identify the appropriate models and parameters. The manuscript is clearly written and describes the modeling process, choice of models, parameter variations, the choice of the model best fitting the data, study conclusions and limitations. I especially praise the full documentation of the models available on GitHub.

There are a few minor comments:

- The use of Figures is non-sequential, with Figure 1A first, then Figure 2A, then Figure 1B, etc. Figure 3A should be also cited on page 10, when the authors start talking about candidate models.

- What are “model A” and “model B” in Supplemental Table S1? Should be M1 and M2.

- “summary data were extracted from published figures using the Figure Calibration function in ImageJ” – can you provide the data in the tabular form. Please be more specific about which figure and which panel where used for data extraction.

- Please refer to specific tables/figures in Jung, Horning and Kinghorn studies, and repeat this data in supplementary material, with captions stating what was omitted.

- What is “custom cost function” on page 13? Please remind the user of the values of n and k for each M1-M8 models.

- Figure 3B – would be useful to name columns M1-M8.

- Page 24: “we chemically inhibited each parameter individually by 90% and investigated how inhibition affects intracellular and extracellular sFLT1 over time” – perhaps the authors meant to say they simulated inhibition of each individual parameter by 90%.

- Page 25: “lysosomal degradation inhibitor chloroquine” and "the Golgi inhibitor brefeldin” – it’s the first time these inhibitors are introduced, authors need to put them into the context of models M1-M8.

Reviewer #2: This paper describes the development and analysis of a computational model of the secretion dynamics of a soluble form of the VEGF receptor called sFLT1. Data from 5 different experimental studies of the process covering two different experimental scenarios, constitutive and pulse-chase, are integrated into a single calibration set that is used to calibrate and evaluate a family of computational models that contain a set of six common processes and encompassing zero to three additional mechanisms, with the goal of identifying a minimal model that best captures the data. This goal is accomplished by comparing each best fit model using the corrected form of Akaike's Information Criterion (AIC_c), which chooses a model that includes only one of the three proposed additional mechanisms, maturation delay, which is modeled with the addition of a single time delay parameter using delayed differential equations. The paper presents a number of additional analyses of the selected model, including parameter identifiability and sensitivity analysis, which is extended to explore possible mechanisms for chemical and genetic perturbation of the system. A key result of this analysis is that while some of these effects are well-predicted by the model, others, notably those that are sensitive to intracellular trafficking dynamics, are not. These results provide a strong rationale for future development of models, presumably based on an expanded set of experimental measurements, that more accurately capture these dynamics. Overall, the paper is well-written and clear, and I have just minor comments that I think may improve the manuscript if they are addressed prior to publication. I congratulate the authors on their thorough and rigorous treatment of this system.

Minor comments

A general comment is that there are several instances of statements where factual statements are made without a corresponding reference. Examples include:

- p. 27: "...the targets of inhibition can affect the trafficking of many proteins, which may themselves have a role in sFLT1 trafficking."

- p. 34: " Importantly, the chemical and genetic inhibitors explored are not specific to sFLT1, but regulate trafficking for many other proteins, some of which may themselves modulate sFLT1 synthesis and secretion."

- p. 36: " Even in monolayer culture of single cell lines, cells are not identical, as the stochasticity of biochemical reactions leads to natural gradients and fluctuations."

p. 13. There is a typo in Eq. 5: log is repeated in the first term of the equation. Also, it is more standard to provide the AICc formula as an equation, in other words to write: "AIC_c = n \log (C/n) + 2k + ..." instead of just giving the formula. (Note also that C should be capitalized in the formula.)

p. 13. " is the cost of a given model calculated using the custom cost function." I think it would be helpful to have an explicit formula for C, referred to in Eq. 5., especially because of the use of the word "custom" here, the reader may think they have missed something.

p. 13. In describing use of AIC_c it should be mentioned that there is an additional assumption here that the variance is constant over all the experimental data points and that the additional model parameter is the value of the variance.

p. 19. In Figure 3C labels M4 and M7 are switched.

p. 24. "...while extracellular sFLT1 initially increases before declining to a new steady state within 48h." This trend cannot be seen on the graph - maybe show as an inset or in a separate graph.

p. 30ff. In the Discussion section it would be helpful to refer to specific figures when reviewing the key results of the paper that are being discussed to increase specificity of the points being made. For example, "A key first step in our study was comparing different mechanistic models for sFLT1

secretion [(Fig 2A and 3A)]." "As a result, model M2 - a DDE-based model that included maturation delay but not internalization or

production decay - was identified as the best model and was used for the remainder of our study [(Fig 3B,C)]." etc.

Fig. 3C - in the combined plot of Cost and AIC there is overlap of the y-axis tick labels for the top and bottom panels. It would help to remove one of the values

Fig. 5D, caption. dX/dt = f(prod)-f(secr) - f(ideg), should be dI/dt.

Fig. 5E, caption, duplicates 5D caption - shows component fluxes not total.

**Have the authors made all data and (if applicable) computational code underlying the findings in their manuscript fully available?**

Reviewer #1: Yes

Reviewer #2: Yes

PLOS authors have the option to publish the peer review history of their article (what does this mean? ). If published, this will include your full peer review and any attached files.

**Do you want your identity to be public for this peer review?** For information about this choice, including consent withdrawal, please see our Privacy Policy .

Reviewer #1: No

Reviewer #2: No

**Figure resubmission:**
---

## [Decision Letter · Decision Letter 1]

10 Jul 2025

Dear Gill,

We are pleased to inform you that your manuscript 'Mechanistic computational modeling of sFLT1 secretion dynamics' has been provisionally accepted for publication in PLOS Computational Biology.

Best regards,

Mariko Okada, PhD

Academic Editor

PLOS Computational Biology

Pedro Mendes

Section Editor

PLOS Computational Biology

Reviewer's Responses to Questions

**Comments to the Authors:**

Reviewer #1: The authors addressed all revisions properly.

Reviewer #2: The authors have thoroughly addressed the critiques raised by both reviewers.

**Have the authors made all data and (if applicable) computational code underlying the findings in their manuscript fully available?**

Reviewer #1: Yes

Reviewer #2: Yes

PLOS authors have the option to publish the peer review history of their article (what does this mean? ). If published, this will include your full peer review and any attached files.

**Do you want your identity to be public for this peer review?** For information about this choice, including consent withdrawal, please see our Privacy Policy .

Reviewer #1: No

Reviewer #2: No

---

## [Editor Report · Acceptance letter]

PCOMPBIOL-D-25-00322R1

Mechanistic computational modeling of sFLT1 secretion dynamics

Dear Dr Gill,

I am pleased to inform you that your manuscript has been formally accepted for publication in PLOS Computational Biology. Your manuscript is now with our production department and you will be notified of the publication date in due course.

With kind regards,

Zsofia Freund
